

# Snow profile alignment and similarity assessment for aggregating, clustering, and evaluating of snowpack model output for avalanche forecasting

Florian Herla[1], Simon Horton[1, 2], Patrick Mair[3], and Pascal Haegeli[1]

[1]Simon Fraser University, Burnaby, BC, Canada
[2]Avalanche Canada, Revelstoke, BC, Canada
[3]Harvard University, Cambridge, MA, USA

**Correspondence:** Florian Herla (fherla@sfu.ca)

**Abstract.** Snowpack models simulate the evolution of the snow stratigraphy based on meteorological inputs and have the potential to support avalanche risk management operations with complementary information relevant to their avalanche hazard assessment, especially in data-sparse regions or at times of unfavorable weather and hazard conditions. However, the adoption of snowpack models in operational avalanche forecasting has been limited, predominantly due to missing data processing

algorithms and uncertainty around model validity. Thus, to enhance the usefulness of snowpack models for the avalanche industry, numerical methods are required that evaluate and summarize snowpack model output in accessible and relevant ways. We present algorithms that compare and assess generic snowpack data from both human observations and models. Our approach exploits Dynamic Time Warping, a well-established method in the data sciences, to match layers between snow profiles and thereby align them. The similarity of the aligned profiles is then evaluated by our independent similarity measure

based on characteristics relevant for avalanche hazard assessment. Since our methods provide the necessary quantitative link to data clustering and aggregating methods, we demonstrate how snowpack model output can be grouped and summarized according to similar hazard conditions. Through emulating a human avalanche hazard assessment approach, our methods aim to promote the operational application of snowpack models so that avalanche forecasters can begin to build understanding in how to interpret and when to trust operational snowpack simulations.

## 1 Introduction

Snow avalanches are a serious mountain hazard, whose risk is managed through a combination of long- and short-term mitigation measures, depending on the character of the exposed elements-at-risk. Avalanche forecasting—the prediction of avalanche hazard over a specific area of terrain (Campbell et al., 2016)—is a critical prerequisite for choosing effective short-term mitiga-

20 tion measures and timing them properly (e.g., publication of advisories, temporary closures, proactive triggering of avalanches).





The task of avalanche forecasters[1] is to integrate the available weather, snowpack and avalanche observations into a coherent mental model of the hazard conditions across their area of interest (LaChapelle, 1966, 1980; McClung, 2002). Statham et al. (2018) describe the essence of avalanche forecasting as answering four sequential questions: 1) *What* type of avalanche problem(s) exist? 2) *Where* are these problems located in the terrain? 3) *How likely* are avalanches to occur? and 4) *How big* will

these avalanches be? Because of the complexity of the avalanche system and the large uncertainty due to the spatial and temporal variability of snowpack properties (Schweizer et al., 2007), avalanche forecasts are subjective expert judgments that are expressed in qualitative degrees of belief (Vick, 2002).

Snow profiles describing the stratigraphy of the snowpack and the characteristics of the individual layers (McClung and Schaerer, 2006) are an important source of information for avalanche forecasting. While avalanche observations offer direct

evidence of unstable conditions, the information on structural weaknesses and slab properties contained in snow profiles is crucial for developing a more complete understanding of the nature of the avalanche hazard, its spatial distribution, as well as making predictions about the likelihood of avalanches and their expected size. To adequately capture the conditions in an area of interest, it is common practice for avalanche forecasters to collect snow profile information from a variety of informative locations and employ targeted sampling to address specific hypotheses. When observing a snow profile, the most commonly

recorded layer characteristics are snow grain type, grain size and layer hardness. In addition, layers representing critical structural weaknesses are often labeled with their burial date to facilitate tracking and simplify communication (Canadian Avalanche Association, 2016). Snowpack tests might be performed to examine the potential for fracture initiation and failure propagation along specific layers of interest.

As a winter progresses, avalanche forecasters continuously synthesize the collected snow profile information into a com-

prehensive picture of existing hazard conditions across the terrain. While experienced forecasters can process snow profile information intuitively and effortlessly, the process is actually a challenging exercise in multi-dimensional pattern recognition, pattern matching, and data assimilation, which requires several advanced skills. These include matching key features between profiles, assessing the similarity or dissimilarity of profiles, combining the information of several profiles into an overall perspective, as well as extrapolating the identified patterns across terrain based on knowledge of snowpack processes and how they

are affected by terrain. In North America, it is common practice among avalanche forecasters to document their understanding of the local snowpack by sketching synthesized snow profiles for different areas of interest (e.g., elevation or aspect specific).

Since the late 1980s, physically-based numerical snowpack models have been developed to expand the available information sources for avalanche forecasters beyond traditional field observations. The most commonly used snowpack models are Crocus (Brun et al., 1989; Vionnet et al., 2012), and SNOWPACK (Lehning et al., 1999; Bartelt et al., 2002; Lehning et al., 2002b, a).

Both of these models simulate the stratigraphy of the snowpack at a point location by integrating meteorological input data over a winter season. The source for the meteorological forcing can be time series of in-situ observations, outputs of numerical weather prediction models, or assimilation products that integrate both. The physical properties of the individual snow layers

---

[1]We use the term avalanche forecaster to describe anybody who assesses avalanche hazard conditions to make decisions about short-term mitigation options. This can include public avalanche forecasters, avalanche safety technicians, ski patrollers, mountain guides, and private recreationists.





(e.g., grain type, grain size, hardness) are simulated using empirical representations of the key snowpack processes (e.g., snow metamorphism, water percolation, settlement) that are tied together by conservation of mass and energy.

Over the last 20 years, extensive amounts of research have been conducted to improve the capabilities of snowpack models, and explore their application for avalanche forecasting. Many contributions evaluated or improved the skill of the models with

respect to hazardous weak layer formation (Fierz, 1998; Bellaire et al., 2011; Bellaire and Jamieson, 2013b; Horton et al., 2014, 2015; Horton and Jamieson, 2016; Van Peursem et al., 2016), or weak layer detection (Monti et al., 2014a). Others tried to assess snow stability from model outputs (Schweizer et al., 2006; Schirmer et al., 2010; Monti et al., 2014b) and estimate danger levels (Lehning et al., 2004; Schirmer et al., 2009; Bellaire and Jamieson, 2013a). Vionnet et al. (2016) and Vionnet et al. (2018) specifically evaluated and improved meteorological data from a weather prediction model serving as input to a

snow cover model. While all studies agree that snowpack modeling has the potential to add value to avalanche forecasting, the understanding of under which circumstances and to which degree these models can add value (especially when coupled with weather prediction models) seems to be limited. This knowledge gap is a major hurdle for developing necessary trust for the operational use of these models.

In Canada, the combination of numerical weather and snowpack models offers a tremendous opportunity for providing

avalanche forecasters with useful information on snowpack conditions in otherwise data-spare regions (Storm, 2012). However, the integration of physical snowpack models into operational avalanche forecasting has so far been limited. Informal conversations with forecasters highlight two main issues: 1) the overwhelming volume of data produced by the models, and 2) validity concerns due to cumulative impact of potentially inaccurate weather inputs. Morin et al. (2020) provide a more detailed discussion of the challenges around the operational use of snowpack models, which the authors classify into four main

categories: issues of accessibility, interpretability, relevance and integrity.

Addressing the two issues effectively requires the development of computer-based methods that can process large numbers of snow profiles. An algorithm for objectively assessing the similarity of simulated snow profiles is the necessary foundation for computationally emulating the snowpack data synthesis process of avalanche forecasters and meaningfully reducing the data volume to a manageable level. Furthermore, the ability to operationally compare simulated snow profiles against observed

ones provides an avenue for continuously monitoring the quality of the simulations and correct them if necessary.

While numerical methods for comparing simulated snow profiles exist, they are unable to address the operational needs described above. To evaluate the performance of SNOWPACK, Lehning et al. (2001) developed an algorithm for comparing modeled profiles against manual observations. Since their approach is only concerned with finding manually observed layers in a specific depth range of the modeled profile, it is not suitable for subsequent clustering and aggregating. Moreover, their

agreement score for snow profile pairs is focused on providing insight for model improvements, which has different similarity assessment needs than comparing snow profiles for avalanche hazard assessment purposes. Hagenmuller and Pilloix (2016) and Hagenmuller (2018) were the first to align, cluster and aggregate one-dimensional snow hardness profiles from ram resistance field measurements using Dynamic Time Warping (DTW), a method from the fields of time series analysis and data mining. While their contribution demonstrates the usefulness of DTW for snow profile comparisons, their method is not general enough

to allow for meaningful comparisons of snow profiles from different sources and varying levels of details.



The objective of this study is to introduce a new approach for computationally comparing, grouping and summarizing snow profiles that can handle both simulated profiles and manual observations with different levels of detail. To maximize the value for avalanche forecasting, our methods focus on structural elements in the profiles that are particularly important for avalanche hazard assessments. We approach the task by numerically emulating the cognitive process of human forecasters: We first

present a layer matching algorithm that aligns profiles in a way that a similarity measure can evaluate their agreement. We then exploit the resulting similarity score between pairs of snow profiles to cluster snow profiles into distinct groups and aggregate them into a representative profile. The derivation of the snow profile alignment algorithm and the similarity measure is presented in Sect. 2, whereas the new methods are valuated through practical aggregation and clustering applications as described in Sect. 3. We discuss the implications of our approach alongside its limitations, and conclude with future perspectives in

Sect. 4. We believe that the algorithms presented in this paper provide an important step for the development of operational data aggregation and validation algorithms that can make large-scale snowpack simulations more accessible and relevant for avalanche forecasters.

## 2 Derivation of the snow profile alignment algorithm and similarity measure

In this section we describe how snow profiles can be aligned by matching layers between them (Sect. 2.2), and define a

similarity measure to evaluate the agreement between aligned profiles (Sect. 2.3). Since both of these tasks require a method for assessing differences between individual snowpack layers, we start with that in Sect. 2.1.

### 2.1 Assessing differences between individual snow layers

To align snow profiles and determine the similarity of snow profiles as a whole, we need a method for assessing the similarity of individual layers. While snowpack models provide a wide range of layer properties, the most commonly used characteristics

by practitioners are snow grain type, layer hardness, and burial date. To assess the similarity between individual layers that incorporate all three layer characteristics, we first define distance functions for these characteristics, which are normalized to the interval $[0, 1]$ to make them comparable. A distance of zero means that two layer characteristics are identical, whereas a distance of one represents complete dissimilarity.

### 2.1.1 Distance function for grain type

The international classification for snow on the ground (Fierz et al., 2009) organizes snow grain types (also known as grain shapes) into main grain type classes, which can in turn be broken down into more nuanced sub-classes. The following grain types are typical in avalanche forecasting contexts: precipitation particles (PP), decomposing and fragmented particles (DF), round grains (RG), faceted crystals (FC) (including the sub-class rounding facets, FCxr), surface and depth hoar (SH and DH), and melt forms (MF) (including the sub-class melt-freeze crusts, MFcr). New snow layers mostly consist of PP and DF, SH

and DH are prototypical persistent weak layers, and MFcr often promote the faceting of adjacent grains, which weakens the interface (Jamieson, 2006). FC, including FCxr, can also be considered persistent weak layers, even though not every faceted



layer is a layer of concern according to common snow profile analysis techniques that consider combinations of grain type and grain size (amongst other properties) to identify structural weaknesses in the snowpack (Schweizer and Jamieson, 2007). Following that concept, we use the the term *bulk layers* for layers that constitute a large proportion of the snowpack without being structurally weak. While avalanche forecasters typically regard FC indicative of weaker snowpack layers, simulated
layers of FC tend to be associated with smaller faceted crystals and thicker layers that represent bulk layers rather than weak layers since SNOWPACK classifies any faceted grain with a size greater than $1.5$ mm as DH.

Since grain type is a categorical variable, calculating distances between non-identical grain types is non-trivial. Our approach builds on the original method developed by Lehning et al. (2001), who defined a matrix of normalized grain type similarities between all possible pairs of grain types based on the physics of their formation and metamorphosis. Their approach evaluates
the modeled grain type stratigraphy to identify model deficiencies and offer insight for model improvements. By contrast, our focus is on matching layers between snow profiles and assessing their similarity for avalanche hazard assessments. For the layer matching task, the similarity between grain types should indeed be evaluated partly based on their formation processes but also on the knowledge of snowpack model quirks and differences between modeled and observed profiles. For assessing the similarity of profiles, however, the similarity between grain types should be evaluated based on their implications for the
hazard conditions rather than their formation processes. Thus, to make the approach more suitable for aligning snow profiles and assessing their similarity, we adapt the grain type similarity matrix of Lehning et al. (2001) for each of the two tasks separately (Table 1A, B). The matrices contain values between $[0, 1]$ that represent the similarity between two grain types $g_1$ and $g_2$. Values $> 0.5$ indicate similarity, $0.5$ implies indifference, and values $< 0.5$ indicate dissimilarity. The similarity between two grain types can be converted into a distance function $d_g(g_1, g_2)$ by subtracting the similarity from 1 (i.e., the
similarity between identical grain types is 1, their distance is 0) to make it comparable to the other distance functions for hardness and layer date. Table 1A and B are modified from the grain type similarity matrix of Lehning et al. (2001) in the following ways (i.e., gray cells):

1. Practical experience with SNOWPACK shows that buried SH layers are often labeled as DH layers. Hence, the original similarity matrix of Lehning et al. (2001) is unable to recognize the similarity of a buried SH layer in a human observed
profile and the equivalent layer in a simulated profile. To address this issue, we raised the similarities between these two grain types from 0 to 0.9 for both tasks (Table 1A, B), despite the differences in their formation process. The increased similarity is also in line with the fact that both SH and DH represent hazardous weak layers and are of comparable importance in avalanche hazard assessments. Following the same line of logic, we also raised the similarity between SH and FC, another common weak layer grain type. However, to acknowledge the enhanced seriousness of buried SH layers
while simultaneously facilitate the weak layer matching, the similarity between SH and FC is higher when aligning snow profiles (Table 1A) than when assessing their similarities (Table 1B)

2. Since human profiles sometimes lack grain type information for certain layers, an automated alignment algorithm needs to be able to cope with missing data in a meaningful way. Since it is more common for bulk layers to have missing grain type information, it is more desirable to match unknown grain types to bulk layers than weak layers. We therefore





**Table 1.** Similarities between snow grain types as used for the layer alignment of snow profiles (A), and as used for the similarity assessment between snow profiles (B). Gray cells highlight modifications to the matrix used in Lehning et al. (2001), and bold fonts highlight differences between the two tables A and B.

**A) Grain type similarity (snow profile alignments)**

|      | PP   | DF   | RG   | FC   | DH   | SH   | MF   | FCxr | MFcr | na     |
|------|------|------|------|------|------|------|------|------|------|--------|
| PP   | 1.00 |      |      |      |      |      |      |      |      | **0.60** |
| DF   | 0.80 | 1.00 |      |      |      |      |      |      |      | **0.60** |
| RG   | 0.50 | 0.80 | 1.00 |      |      |      |      |      |      | **0.60** |
| FC   | 0.20 | 0.40 | 0.40 | 1.00 |      |      |      |      |      | 0.50   |
| DH   | 0.00 | 0.00 | 0.10 | **0.80** | 1.00 |      |      |      |      | **0.40** |
| SH   | 0.00 | 0.00 | 0.00 | **0.60** | 0.90 | 1.00 |      |      |      | **0.40** |
| MF   | 0.00 | 0.00 | 0.00 | 0.00 | 0.00 | 0.00 | 1.00 |      |      | 0.50   |
| FCxr | 0.20 | 0.40 | 0.50 | **0.80** | **0.70** | 0.00 | 0.00 | 1.00 |      | **0.60** |
| MFcr | 0.00 | 0.00 | 0.00 | 0.00 | 0.00 | 0.00 | 0.20 | 0.00 | 1.00 | **0.40** |

**B) Grain type similarity (snow profile similarity assessments)**

|      | PP   | DF   | RG   | FC   | DH   | SH   | MF   | FCxr | MFcr | na     |
|------|------|------|------|------|------|------|------|------|------|--------|
| PP   | 1.00 |      |      |      |      |      |      |      |      | **0.50** |
| DF   | 0.80 | 1.00 |      |      |      |      |      |      |      | **0.50** |
| RG   | 0.50 | 0.80 | 1.00 |      |      |      |      |      |      | **0.50** |
| FC   | 0.20 | 0.40 | 0.40 | 1.00 |      |      |      |      |      | 0.50   |
| DH   | 0.00 | 0.00 | 0.10 | **0.50** | 1.00 |      |      |      |      | **0.50** |
| SH   | 0.00 | 0.00 | 0.00 | **0.30** | 0.90 | 1.00 |      |      |      | **0.50** |
| MF   | 0.00 | 0.00 | 0.00 | 0.00 | 0.00 | 0.00 | 1.00 |      |      | 0.50   |
| FCxr | 0.20 | 0.40 | 0.50 | **0.60** | **0.40** | 0.00 | 0.00 | 1.00 |      | **0.50** |
| MFcr | 0.00 | 0.00 | 0.00 | 0.00 | 0.00 | 0.00 | 0.20 | 0.00 | 1.00 | **0.50** |

expanded the similarity matrix of Lehning et al. (2001) with an additional column for unknown grain types and filled it with values that are centered around indifference (i.e., 0.5) (Table 1B), but with slight preference towards bulk layers (Table 1A).

3. While we acknowledge the physical similarity between DH, FC, and FCxr, we want to emphasize their slightly different implications for the hazard conditions. We therefore classified DH layers—often representing buried SH layers, or layers with large FC grains in simulated profiles—as first order weak layers, whereas FC and FCxr were classified as second and third order weak layers, respectively. This hierarchy is implemented in Table 1B: similarities DH–FC and DH–FCxr are equal to and slightly lower than 0.5, respectively, and thus represent indifference or a slight mis-match; the similarity FC–





FCxr is slightly greater than $0.5$ and thus represents a weak match. These modifications lead to a pronounced distinction between the weak layer grains SH and DH from the less distinct weak layer grains FC and FCxr.

### 2.1.2 Distance function for layer hardness

The second important layer characteristic to consider is hardness, which characterizes the resistance of snow to penetration. In operational field observations the layer hardness is expressed on an ordinal hand hardness scale (Fierz et al., 2009). Hardness observations are taken by gently pushing different objects into snow layers from a snow pit wall. The hardness of a layer is expressed by the largest object that can be pushed into the snow layer with a consistent force of approx. 10–15 N. The ordinal levels of the hand hardness index are fist (F), four fingers (4F), one finger (1F), pencil (P), knife blade (K), and ice (I). Sub-classications like *4F+, 4F-1F, 1F-* are possible and refer to *just harder than 4F, between 4F and 1F*, and *just softer than 1F*, respectively.

A translation of the ordinal index into a numerical scale is straightforward by assuming that *fist* equals to a numerical value of 0, *ice* equals to 6, and the ordinal levels are equidistant (Schweizer and Jamieson, 2007). The normalized distance $d_h$ can then be written as $d_h(h_1, h_2) = \frac{|h_1 - h_2|}{5}$, where $h_1$ and $h_2$ represent numerical translations of two hand hardness values and the normalization factor of $5$ refers to the largest distance possible (F–I). Hence, only layers with hardness difference of fist to ice are considered completely dissimilar, while all other hardness combinations exhibit some degree of similarity.

### 2.1.3 Distance function for layer date

Snow layers are commonly labeled with either their deposition date (i.e., the date when a specific layer was formed) or their burial date (i.e., the date when a specific layer was buried). While snowpack models predominantly work with deposition dates, practitioners mainly use burial dates. One reason for practitioners' preference for burial dates is the fact that layers can form over several days, which makes assigning a deposition date challenging. However, it is straightforward to derive the burial date of a simulated snowpack layer based on the deposition date of the overlying layer. Hence, layer dates of simulated snow profiles can easily be compared with layer dates recorded by practitioners.

The distance function $d_t$ between two dates $t_1$ and $t_2$ becomes trivial as soon as the dates represent the same type of date (deposition or burial) and are converted into Julian dates: $d_t(t_1, t_2) = \frac{|t_1 - t_2|}{c}$. In this case, the normalization factor $c$ determines the time lag when the dates are considered to be completely dissimilar. A normalization factor of $c = 5$, for example, means that date differences from zero to four days become increasingly dissimilar (i.e., $d_t \in [0, 1]$), whereas date differences equal to or greater than five are considered completely dissimilar (i.e., $d_t \geq 1$; $d_t$ typically does not exceed 2, but the exact limit depends on $c$ and the length of the season or the size of the DTW warping window constraint). A normalization factor of $c = 1$ means that only identical dates are considered to have any similarity. Hence, $c$ can be used to account for small deviations in reported burial dates as well as in short time lags of weather patterns across geographic regions. In cases when layer dates are not available for one or both layers, the distance $d_t$ defaults to *indifference* (i.e. $d_t = 0.5$). This makes it possible to label only important layers with their date.





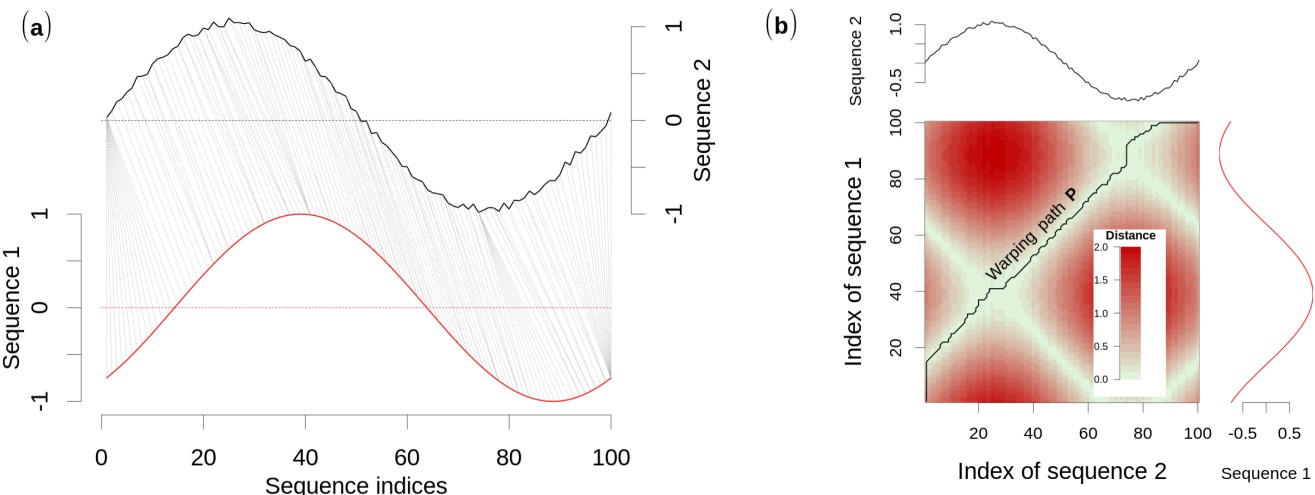

**Figure 1.** Illustrative example of the DTW alignment of two sine wave sequences. (a) each element of the first sequence is mapped to one or many elements of the second sequence; the corresponding sequence elements can be found by (b) the (warping) path of least resistance through a local cost matrix that stores the the distances between every possible pair of sequence elements.

## 2.2 Aligning snow profiles with Dynamic Time Warping

In this section, we present a numerical algorithm that addresses the challenge of matching corresponding layers in snow profiles. Our method is based on Dynamic Time Warping (DTW), a longstanding algorithm, which was originally designed for speech recognition in the 1970s (Sakoe and Chiba, 1970; Sakoe, 1971; Sakoe and Chiba, 1978). Soon thereafter it was

adopted in time series analyses (e.g., Berndt and Clifford, 1994), and it remains a state-of-the-art component in data mining methods such as clustering and classification (e.g., Keogh and Ratanamahatana, 2005; Petitjean et al., 2011; Wang et al., 2013; Paparrizos and Gravano, 2015). Our discussion of DTW starts with a general background section, which is followed by four sections that explain our application of DTW to snow profiles in more details. This includes snow profile pre-processing steps as well as inputs to the DTW algorithm. We then discuss suitable DTW parameter choices for snow profile alignments, and

make recommendations on how to use the alignment algorithm.

### 2.2.1 Background on Dynamic Time Warping

The following brief summary of DTW is based on Sakoe and Chiba (1978), Rabiner and Juang (1993), Keogh and Ratanama-hatana (2005), and Giorgino (2009).

DTW is an *elastic* distance measure for time series or, more generally, sequences that calculates the dissimilarity between

two sequences while allowing for distortions in "time". The distortions are accommodated by mapping each element of one sequence to one or many elements of the other sequence (Fig. 1a). Once mapped, the dissimilarities between each of the





matched sequence elements can be computed and combined into one single distance value that quantifies the dissimilarity between the two sequences.

To link corresponding sequence elements, a local cost matrix $\mathbf{D}$ is calculated that stores the distances between every possible pair of sequence elements. Hence, the dimensions of $\mathbf{D}$ are the lengths of the two sequences. Then, the best alignment of the
two sequences is represented by the path of least resistance through $\mathbf{D}$, which is referred to as the optimal warping path $\boldsymbol{P}$ (Fig. 1b). To ensure meaningful matching between sequence elements, the warping path is subject to the following constraints:

**Monotonicity and continuity** Subsequent elements of the warping path are contiguous in a sense, that they are horizontally, vertically or diagonally adjacent cells of the local cost matrix $\mathbf{D}$, while no "stopping" or "going back" is allowed.

**Warping window** It is common practice to restrict the search for the optimal warping path to the bounds of a warping window
around the main diagonal of the local cost matrix $\mathbf{D}$. That main diagonal represents the lock-step alignment of the two sequences, where the $i$th element of the first sequence is mapped onto the $i$th element of the second sequence. Thus, the farther the warping path deviates from the main diagonal, the more extreme the warping gets, and sequence elements that are farther apart from each other get matched. To prevent excessive warping, the warping path is often constrained to a warping window. The shape and size of the optimal warping window changes with the domain and the target data set.
A slanted band of constant width (*Sakoe–Chiba band*) or a parallelogram between the sequences' start and end points (*Itakura parallelogram*) are popular choices. For detailed visualizations see Ratanamahatana and Keogh (2004). DTW with a constrained warping window is commonly referred to as *constrained* Dynamic Time Warping (cDTW).

**Local slope constraint** While the warping window constrains the envelope of the warping path *globally*, the so-called local slope constraint of the warping path ensures reasonable warping *locally*. More specifically, the local slope constraint
controls how many subsequent elements of one sequence can be mapped onto one element of the other sequence. That is an important control, because it regulates how much stretching and compressing of individual sequence elements is allowed. For time series, that means stretching and compressing with respect to time; in the case of snow profiles, it refers to the stretching and compressing of snow layer thicknesses.

Many different local slope constraints have been suggested in the literature. In Fig. 1, we use a non-restrictive local slope
constraint, which allows arbitrarily many elements to be mapped onto one corresponding element. In this case, the first element of sequence 2 is mapped onto almost 20 elements of sequence 1 (Fig. 1a), which requires a vertical start slope of the warping path (Fig. 1b).

**Boundary conditions** For a *global* alignment, both the sequences' start and end points need to be elements of the warping path (Fig. 1b). *Partial* alignments can be computed by relaxing one or both of these constraints. This results in three different
options: alignments where the start but not the end points are matched (*open-end* alignment), alignments where the end but not the start points are matched (*open-begin* alignment), or alignments where subsequences of the two sequences that neither include the start or end points are matched. Further details on partial DTW matching can be found in Appendix A and in the review by Tormene et al. (2009).





While there are many potential warping paths through the local cost matrix **D** that satisfy these constraints, the objective is to find the *optimal* warping path $P$ that accumulates the least cost while stepping through **D**. This is an optimization problem that can be solved by dynamic programming. We refer the interested reader to Appendix A for more details.

### 2.2.2 Preprocessing of snow profiles: uniform scaling and resampling

The specific nature of snow profile data requires some preprocessing before DTW can be applied in a meaningful way. Variabilities in snowpack structures can be divided into systematic differences due to systematic variations in the meteorological forcing (e.g., location: a wind scoured ridgeline next to a wind loaded slope; elevation: increase of snowfall amounts with elevation), and random differences due to natural variations in the meteorological forcing (e.g., peculiar patterns of individual storms; small-scale variations) (Schweizer et al., 2007). The former can result in substantial differences in the snow profiles due

to accumulation of different forcings over time. (See Sect. S1 in the supplementary material for more details and visualizations of idealized stratigraphic snowpack variability.) While the DTW algorithm is well suited to deal with random differences, it was not designed to cope with systematic differences. Since systematic differences are common in snow profiles, it is necessary to preprocess them for a meaningful application of DTW. Fu et al. (2007) suggest Uniform Scaling—another optimization technique—which minimizes systematic differences by determining an optimal, global scaling factor. For efficiency reasons,

we simply scale the snow profiles to identical snow heights instead. We thereby assume that the offset corresponds to approximately the magnitude of the systematic differences, and that the rescaled profiles are characterized predominantly by random differences that can be handled by DTW. A tentative evaluation of this assumption can be found in the supplementary material (Sect. S2.2).

  Once the profiles have been rescaled, each of the two profiles consists of a series of discrete layers along an irregular height

grid. To equalize the two different height grids, we resample the profiles onto a regular grid with a constant sampling rate, which represents the final resolution for the alignment procedure. A typical sampling rate might be of the order of half a centimeter to capture hazardous, thin weak layers. To preserve the discrete layer character of each profile, we do *not* interpolate between the grid points during the resampling process.

### 2.2.3 Computing a weighted local cost matrix from multiple layer characteristics

Section 2.2.1 introduced how DTW exploits the local cost matrix **D** to find the best alignment of two sequences. We will now show how to compute this local cost matrix for snow profiles. Therefore, the following presentation focuses on the layer characteristics of categorical grain type, ordinal layer hardness, and numerical layer date, which are the layer characteristics most commonly recorded by practitioners. Note, however, that the layer date contribution can be omitted if the information is not available, and our algorithm can easily be expanded to include other layer properties.

First, we combine the distances of the individual layer characteristics (defined in Sect. 2.1) into one scalar distance by weighted averaging. The resulting scalar distance fills one cell of the local cost matrix **D**, and thus controls how the alignment algorithm matches the corresponding layers in the snow profiles.



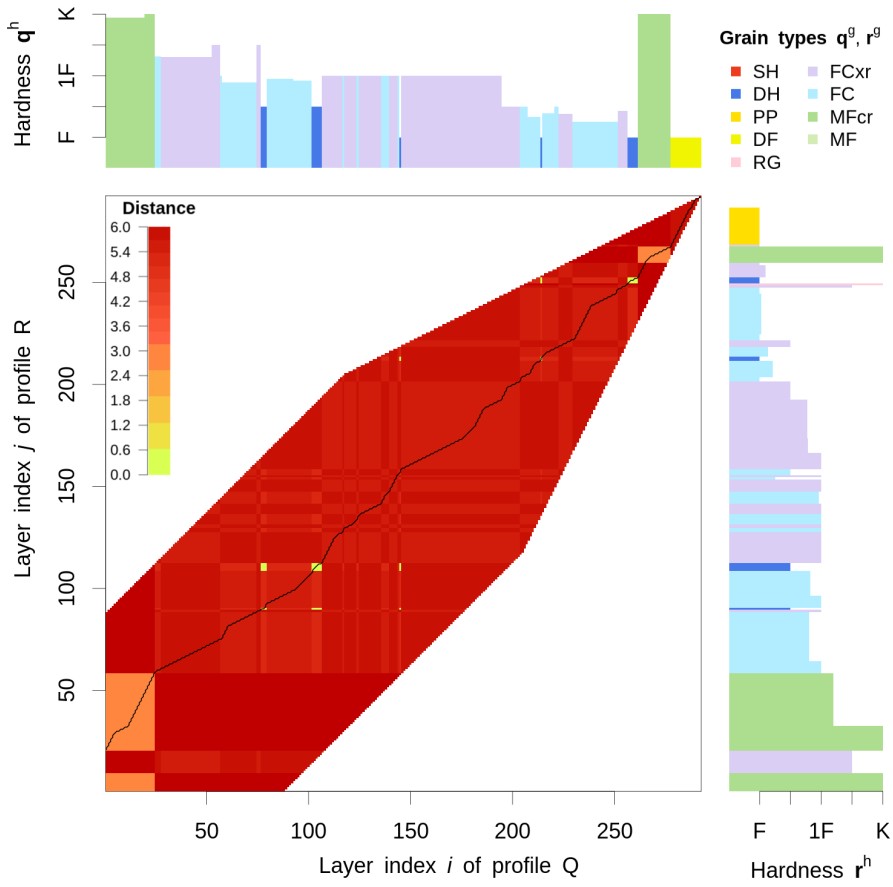

**Figure 2.** Visualization of a local cost matrix **D**, which stores the distance between individual layers of two snow profiles Q and R; Q and R contain the layer characteristics grain type $q^g$ and $r^g$, as well as hardness $q^h$ and $r^h$. **D** is only filled with values around its diagonal, the rest is clipped by a warping window and a local slope constraint; the back line represents the optimal warping path **P**, which accumulates the least cost while stepping through **D**, and which defines the alignment of the two profiles; the preferential layer matching implementation becomes apparent in the yellow and orange cells, which represent smaller than average distances and thus will be matched more easily; see text for further explanation.

Second, we use a weighting scheme for preferential layer matching (Fig. 2) to ensure that the algorithm prioritizes the alignment of snowpack features that are relevant for avalanche hazard assessment when calculating the scalar distance. The intent of the weighting scheme is to create anchor points for key layers by artificially introducing penalties for non-key layers. This is especially advantageous when no date information is available. Similar to our approach with the distance function for grain types, we focus on avalanche hazard assessment priorities and try to emulate a human alignment approach: First priority should be given to the alignment of first-order persistent weak layers (SH and DH), followed by crusts. Second priority should be given to the alignment of faceted grain types with first-order persistent weak layers. This cascade of priorities is





**Table 2.** Weighting scheme for preferential layer matching; the preferential layer matching coefficient $\nu(g_1, g_2)$ depends on the combination of two grain types $g_1$ and $g_2$; the darker the shading, the more preferably the grain type combinations get matched. Values in the upper triangle are symmetric to values in the lower triangle.

|      | PP | DF | RG | FC  | DH | SH | MF | FCxr | MFcr |
|------|----|----|----|-----|----|----|----|------|------|
| PP   | 5  |    |    |     |    |    |    |      |      |
| DF   | 5  | 5  |    |     |    |    |    |      |      |
| RG   | 5  | 5  | 5  |     |    |    |    |      |      |
| FC   | 5  | 5  | 5  | 5   |    |    |    |      |      |
| DH   | 5  | 5  | 5  | 4.5 | 0  |    |    |      |      |
| SH   | 5  | 5  | 5  | 4.5 | 0  | 0  |    |      |      |
| MF   | 5  | 5  | 5  | 5   | 5  | 5  | 5  |      |      |
| FCxr | 5  | 5  | 5  | 5   | 5  | 5  | 5  | 5    |      |
| MFcr | 5  | 5  | 5  | 5   | 5  | 5  | 5  | 5    | 2.5  |

**Table 3.** An illustrative example of a resampled snow profile Q that contains information about the vertical position of the profile layers (at their top interfaces), the grain types $\boldsymbol{q}^g$, the hardnesses $\boldsymbol{q}^h$, and the (burial) dates $\boldsymbol{q}^t$.

| Height (cm) | $i$ | $\boldsymbol{q}^g$ | $\boldsymbol{q}^h$ | $\boldsymbol{q}^t$ |
|-------------|-----|------|------|------------|
| 0.5         | 1   | DH   | 1.0  | 2018-12-15 |
| 1.0         | 2   | DH   | 1.0  | 2018-12-15 |
| $\vdots$    |     |      |      |            |
| 119.5       | 239 | DF   | 2.25 | —          |
| 120.0       | 240 | PP   | 1.5  | —          |

expressed numerically by the relative differences of the weighting coefficients $\nu$ presented in Table 2. We determined these values experimentally by testing numerous snow profile alignments and found that they yield the wanted result without any unwanted side effects.

To compute **D** from two generic snow profiles, we introduce the following notation. Q and R are two rescaled and resampled
5  snow profiles with $I$ number of layers—typically on the order of hundreds of layers, depending on the sampling rate and the time of the season. The indices $i = 1, ..., I$ and $j = 1, ..., I$ refer to these layers. Each layer of the snow profiles Q and R contains information about the grain type, the hardness, the burial date, and the vertical position of the layer in the profile (i.e., height or depth). Those characteristics are denoted by $\boldsymbol{q}^g$, $\boldsymbol{r}^g$ for grain type; $\boldsymbol{q}^h$, $\boldsymbol{r}^h$ for hardness; and $\boldsymbol{q}^t$, $\boldsymbol{r}^t$ for date (see Table 3 for an example). Each element $\mathrm{D}_{ij}$ of the local cost matrix **D** can then be written as

10  $$\mathrm{D}_{ij} = w_g d_g(q_i^g, r_j^g) + w_h d_h(q_i^h, r_j^h) + w_t d_t(q_i^t, r_j^t) + \nu(q_i^g, r_j^g), \tag{1}$$





where $w_g$, $w_h$, and $w_t$ are averaging weights that sum up to 1 ($w_g + w_h + w_t = 1$). Those weights need to be estimated (see supplementary material, Sect. S2.1), but specific values are recommended in Sect. 2.2.5. Since $d_g$ and $d_h$ range within $[0,1]$, $d_t$ typically within $[0,2]$, and $\nu$ within $[0,5]$, the distance $D_{ij}$ typically ranges within $[0,7]$, where a distance of zero ($D_{ij} = 0$) refers to the two layers being identical. Note that—in Equ. (1)—$d_g$ is calculated based on the similarity matrix in Table 1A,

which is geared towards snow profile alignments. Figure 2 visualizes a local cost matrix derived from two generic snow profiles.

### 2.2.4   Obtaining the optimal alignment of the snow profiles

After calculating the local cost matrix $\mathbf{D}$, there are several constraints on the warping path that need to be specified to tailor DTW to snow profile alignments; those constraints are the warping window, the local slope constraint, and the boundary conditions.

**Warping window**   We use a slanted band of constant width around the main diagonal of $\mathbf{D}$ to constrain the warping path. This so-called *Sakoe-Chiba* band is quantified by the window size $\varepsilon$. See Sect. 2.2.5 for a recommendation on which value of $\varepsilon$ to use for snow profile alignments.

**Local slope constraint**   We require the local slope constraint to prevent excessive stretching or compressing of the snow layers in either of the two profiles. The *symmetric* Sakoe-Chiba local slope constraint (Sakoe and Chiba, 1978) does exactly

that: it limits the amount of stretching or compressing to a specific factor, which is identical for both profiles. We chose a factor that limits the amount of stretching to double the layer thickness, and the compression to half the layer thickness. On the technical level, that means that while stepping through $\mathbf{D}$, a horizontal or vertical step is only allowed if following a diagonal step (Fig. A1).

Note that the local slope constraint results in a funnel-shaped restriction of the warping window close to the starting point

of the alignment, which is more restrictive than the slanted band window (Fig. 2). See Appendix A for a more detailed explanation.

**Boundary conditions**   In cases when defining features of the snowpack at the very bottom or top only exist in one of the two profiles (e.g., lack of an early season snowfall event or missing of the most recent storm at one of the two profile locations), partial snow profile alignments are more suitable than global alignments. In those situations the alignment

benefits from relaxing the boundary conditions to accommodate those partial alignments (Fig. 2 and 3). We therefore implement symmetric open-end alignments, where an *entire* profile is mapped onto the other profile with the start points matched but the end points not.

Since *open-begin* alignments cannot be calculated with our chosen local slope constraint (Tormene et al., 2009), we developed a workaround by aligning snow profiles both *bottom-up* and *top-down*. The top-down alignment can be cal-

culated straightforwardly by mirroring $\mathbf{D}$ about its anti-diagonal, and re-running the DTW algorithm on the mirrored matrix. However, since the DTW distance (Sect. 2.2.1, Appendix A) cannot be used to effectively identify the better



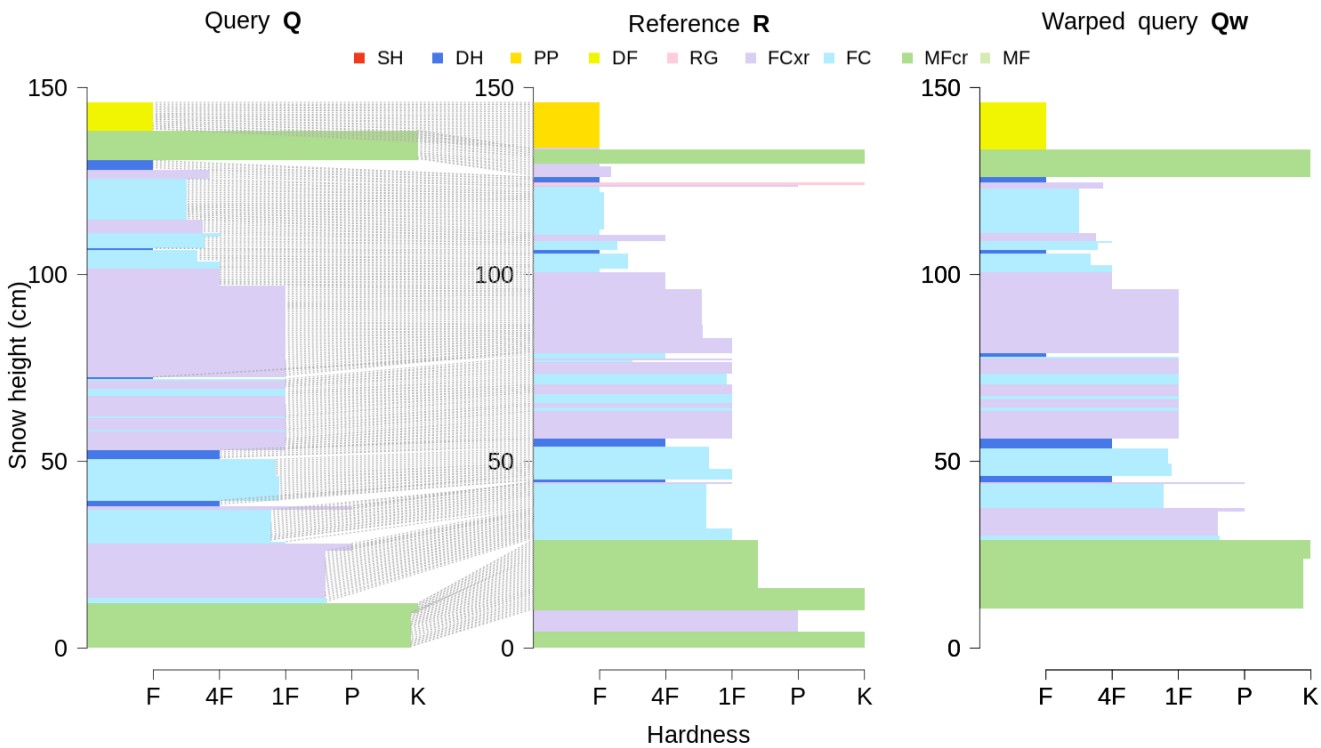

**Figure 3.** The open-begin alignment of two snow profiles Q and R: the line segments match the corresponding layers between Q and R. Adjusting the layer thicknesses of profile Q yields the warped profile $Q_w$, which is optimally aligned to profile R.

alignment of the two, we use our independent, more nuanced similarity measure of snow profiles (addressed in Sect. 2.3) to assess the quality of the alignment.

Finding the optimal warping path $P$ implies matching the corresponding layers between the two profiles Q and R. That in turn allows *warping* one profile onto the other one by optimally stretching and compressing its individual layer thicknesses, so that the *warped profile* is optimally aligned with the other profile. For example, warping profile Q onto profile R produces the warped profile $Q_w$, which contains the same layer sequences as profile Q, but the adjusted layer thicknesses have aligned $Q_w$ to the corresponding layers of R (Fig. 3). The optimally aligned profiles R and $Q_w$ can now be used as input to an independent similarity measure.





### 2.2.5 Application cases and usage recommendations

There are two main application cases for the snow profile alignment algorithm: (i) aligning snow profiles from different locations (or sources) but at the same points in time, and (ii) aligning snow profiles from the same location at different points in time.

In the first case, we recommend to use open-end alignments when the optimal alignment of *all individual* layers is required—e.g., in applications that compare individual layers, or in clustering applications that search for regions of similar snowpack conditions. We recommend to use global alignments when modeled profiles are evaluated versus human profiles: even though open-end alignments might yield better alignment results, the explicit mismatch of layers at the very bottom or top of the profiles represents an important element of the model evaluation. Furthermore, mind that open-end alignments increase the

scope of the alignment algorithm, which can sometimes lead to surprising layer matches if the algorithm is used unsupervised.

In the second case, aligning profiles from the same location at different points in time (i.e., layer tracking), one of the two profiles typically has more layers and a higher snow depth. Since the smaller profile should be contained in the taller profile in a similar form, the alignment needs to be open-end and bottom-up. Furthermore, it is *not* necessary to rescale the profiles as recommended in Sect. 2.2.2, but only to resample them. In this case, rescaling actually moves corresponding layers farther

apart, which would have to be compensated by increasing the window size $\varepsilon$. To increase your control on which parts of the profiles get matched, we recommend that you label some of the key layers with their date information.

In the supplementary material (Sect. S2), we derive the optimal values for the averaging weights $w_g$ and $w_h$, and the window size $\varepsilon$ by simulating experiments. Based on the results of these experiments, we recommend to use the following default settings: $\varepsilon \approx 0.3$ in conjunction with a bottom-up/top-down approach[2]; a ratio of $w_h/w_g = 1/4$; the value of $w_d$ depends

heavily on how similar the meteorological processes are that shape the snowpack at the two locations; e.g. two profiles from the same elevation and the same aspect that are in close proximity can be aligned based on layer date alone; however, two profiles from opposite aspects and different elevation bands, may be aligned predominantly based on grain type and hardness.

### 2.3 Assessing the similarity of snow profiles

A measure that quantifies the similarity of snow profiles as a whole is best designed independently from a profile alignment

or layer matching routine. Such an independent approach allows for *matching of layers based on physical similarity* (i.e., processes like grain formation and metamorphism, or knowledge about snow cover models), whereas *the similarity of the aligned profiles can be assessed based on characteristics relevant for avalanche hazard assessment*. Therefore, we define a similarity measure $\Phi$ for generic snow profiles, based on the layer characteristics grain type and hardness, that can be used to numerically compare, evaluate, and group snow profiles.

Let us again consider two snow profiles. Some of their layers have been matched, while others have not. The non-matched layers are located either at the very bottom or very top of *one* of the two profiles. For example, the two profiles could be the profiles R and $Q_w$ from the previous section. Our goal is to compute a scalar number that expresses the similarity between

---

[2]Note that this value is much higher than typically recommended in the literature (Ratanamahatana and Keogh, 2004).





these two profiles on a scale from 0–1. To do so, we start again by computing the (dis)similarities between the corresponding layers analogously to the previous sections. However, in a similarity measure that is geared towards hazard assessment not every layer is equally important. Furthermore, since important weak layers are often much thinner than the bulk layers, they would be dramatically underrepresented in a measure that computes a standard average across all layers. Thus, we bin all layers

according to four major grain type classes relevant for avalanche hazard assessments: (1) new snow crystals (PP and DF) that are commonly associated with surface problems, (2) weak layers (SH and DH) and (3) crusts (MFcr) that are typically related to persistent avalanche problems, and (4) all other grain types that represent bulk layers. We calculate separate similarity values for every class (a scalar value between $[0, 1]$), and the overall similarity between the two profiles is the average similarity derived from the classes.

To calculate the similarity for a grain type class, we first distinguish between matched and non-matched layers. All non-matched layers are treated as *indifferent*, and are therefore assigned a similarity value of $0.5$. Such a strategy makes the measure robust against a varying number of non-matched layers. Next, we calculate the similarities of all matched layers. That can be done with the distance functions from Sect. 2.1, which compute the dissimilarity between two layers based on grain type or hardness. Note that in this context, the grain type distance is calculated based on Table 1B to ensure the derived similarity is

most useful for avalanche forecasting. The resulting distance is converted into a similarity by subtracting the distance from $1$ (i.e., a distance of $0.8$ becomes a similarity of $0.2$). If the grain type class is new snow crystals or bulk grains, the similarity of a matched layer is computed as the product between the associated similarity of grain type and hardness. The emerging similarity for the entire class is then the average over all (matched and non-matched) layers within.

While the above approach works well for new snow crystals and bulk layers, weak layers and crusts require additional
considerations to be integrated in a meaningful way.

1. Given weak layers or crusts, an identical match of the grain type is arguably more important than a hardness evaluation: many weak layers and crusts are thin, and often melt-freeze crust laminates are characterized by an inhomogeneous hardness. These circumstances challenge precise hardness measurements of these layers and make them prone to error. Additionally, crusts play an important role in avalanches not as a weak layer themselves, but as a layer favoring adjacent
weak layer growth (Jamieson, 2006). That in turn makes the grain type of a crust much more important than its hardness when evaluating the similarity between two layers. In summary, a hardness evaluation might introduce more error than benefit—especially when comparing human versus modeled profiles. Therefore, we compute the similarity of a matched weak layer or crust as the associated similarity of grain type alone, thereby neglecting hardness information.

2. For weak layers and crusts, it is specifically important where in the profile the layers are located. Consider a snow profile
with two DH layers close to the ground and one SH layer buried under new snow. A second, almost identical profile lacks the buried SH layer. While the likelihood of triggering and potential size of avalanches are similar with respect to the two matched DH layers, they are not with respect to the buried SH layer, which is missing in the second profile. Even though the two profiles are visually almost identical, they require different avalanche risk management approaches. If we calculated the similarity for the weak layer class of those two profiles as described above (i.e., as average over all





layers), the thin SH layer would be heavily underrepresented among the thicker DH layers. As a consequence, the weak layer class would exhibit a high similarity value. For example, if the two DH layers are each 10 cm thick, and the SH layer 2 cm, the sampling rate were 1 cm, then the similarity for the weak layer class would be $(20 \cdot 1 + 2 \cdot 0)/22 = 0.9$. Given that the two snowpack conditions demand different risk management approaches, such a high similarity is not meaningful. To mitigate those situations, we divide the two profiles into sections of equal thickness and evaluate the similarity of the weak layers and crusts within those sections separately. The number of sections is determined by the maximum number of weak layers (or crusts, respectively) in either of the two profiles. By evaluating similarities of adjacent weak layers or crusts, we introduce a basic weighting scheme for the position of those layers in the profile. It is based on the idea that avalanche likelihood and size, as well as resulting risk management are rather similar for adjacent weak layers or crusts, but rather different for weak layers or crusts in opposing depths of the profile. In our example, there are three weak layers, hence, the two profiles are divided into three sections of equal thickness. We assume that all weak layers that are in the same section require a similar risk management approach. So, the similarity for each section is the average similarity of the weak layers within, and each section is equally important for the hazard assessment, so the similarity for the weak layer class is the average similarity with respect to the sections. In our example, the lower section contains the two DH layers, the middle section contains no weak layers, and the upper section contains the SH layer in one profile. Hence the similarity for the weak layer class is $\left( \frac{20 \cdot 1}{20} + \frac{2 \cdot 0}{2} \right)/2 = 0.5$. A weak layer similarity of $0.5$ evaluates the two snowpack conditions much better than a similarity of $0.9$.

In summary, the resulting similarity measure $\Phi$ between two snow profiles expresses the similarity between these two profiles in a scalar value within $[0, 1]$. A similarity of $1$ corresponds to the two profiles being identical. Note that the measure is symmetric to the two profiles, so that the first profile is as similar to the second profile as vice versa. By treating non-matched layers as indifferent the measure is able to cope with varying numbers of missing layers at the bottom or top of the profiles, which is important for assessing the similarity of snow profiles from different times and/or locations. To make the similarity measure useful for avalanche forecasting, it weighs hazardous thin layers, crusts, as well as storm snow layers more heavily than bulk layers, and it considers the relative depth of those layers.

## 3 Aggregation and clustering applications—a practical valuation

Sections 2.2 and 2.3 detail how to match layers between snow profiles, and how to assess the similarity of the aligned profiles for applications in avalanche hazard assessment. Both of these steps are fundamental prerequisites to automate forecaster tasks, such as grouping similar profiles and finding the representative profile of a group. In the data sciences, these tasks are called data clustering and data aggregation. In this section, we demonstrate how to apply simple data clustering and data aggregation methods to snow profiles based on their prior alignment and similarity assessment. We use these application examples as face validation of our methods.



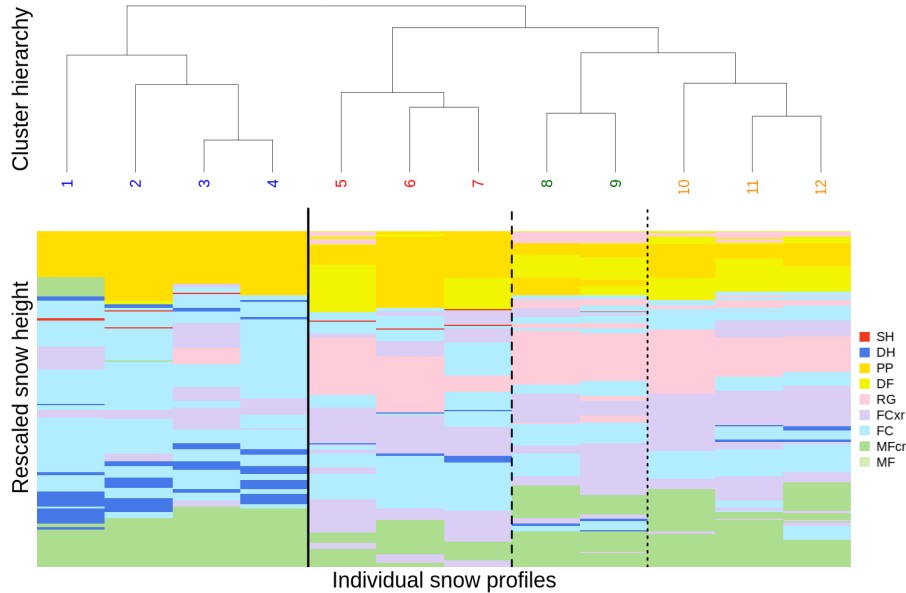

**Figure 4.** Hierarchical clustering of twelve snow profiles based on prior snow profile alignment and similarity assessment. The colors represent the grain types of distinct layers, the bold black lines separate the four most distinct clusters.

## 3.1 Clustering of snow profiles

Avalanche professionals need to group snow profiles to understand how snowpack conditions and avalanche hazard vary across space. The snow profile alignment algorithm and similarity measure described in the previous section enable the automation of this task by providing the necessary quantitative link to the well-established field of numerical clustering methods (e.g., James

et al., 2013; Sarda-Espinosa, 2019).

To showcase the clustering of snow profiles based on our similarity measure, we are using a set of twelve snow profiles that exhibit both pronounced and subtle differences in their snowpack features (Fig. 4). The alignment and similarity assessment of the snow profiles is based on grain type and hardness information, even though they are visualized in Fig. 4 solely by their grain type sequences. After computing a total $12 \cdot 11 = 132$ profile alignments and similarity assessments, the clustering is carried

out by an agglomerative hierarchical clustering algorithm that iteratively fuses individual profiles to clusters based on the similarity of the profiles[3]. The resulting cluster hierarchy is best analyzed from the top to the bottom, allowing to separate the set of twelve profiles into up to twelve clusters. The higher a split occurs in the hierarchy, the more distinct the corresponding clusters are to each other.

The first, most distinct split separates the heavily faceted profiles 1–4 from the remaining profiles. The remaining profiles

are generally quite similar, except for some subtle, but important features that still require different risk management strategies: Profiles 5–7 show weak layers both in the middle of the snowpack as well as below the new snow (i.e., second split); profiles

---

[3]We use complete linkage to fuse individual profiles to clusters (James et al., 2013).





8 and 9 have a weak layer sandwiched between two crusts at the bottom of the snowpack (i.e., third split), and profiles 10–12 either have a weak layer only in the middle of the snowpack or none at all. Additionally to examining those most distinct, four clusters, the similarities within the clusters can be investigated further. For example, profile 1 is the most dissimilar profile within the first cluster, being the only profile with a crust below the new snow and the only profile with a pronounced weak

layer in mid snow height. Or profile 10 is the outsider in cluster 4, having no weak layer at all.

The relationships among the profiles as established by the alignment algorithm and similarity measure yield a sound clustering result that looks similar to how a human avalanche forecaster would group the profiles according to different strategies on how to manage snowpack conditions and the related avalanche hazard. This example demonstrates that our approach can differentiate between both very different as well as subtly different snowpack conditions.

## 10   3.2   Finding a representative snow profile: The medoid

Avalanche forecasters often draw a representative snow profile that summarizes the most important snowpack features within a group of profiles. In the data mining community, this type of generalization is typically called the *average sequence*, or *aggregate*, and the most sophisticated methods for computing that aggregate are closely tied to the used alignment algorithm and similarity measure (e.g., Petitjean et al., 2011; Paparrizos and Gravano, 2015). In the following, we use the simple approach

of identifying the one profile within the group that is most similar to all other profiles, called the *medoid* profile. Visually, the medoid profile can be thought of as the member of the group that is closest to the geometric center of the group. Mathematically, that means that the medoid profile minimizes the accumulated distances to all other profiles. To identify the medoid, we compute the accumulated distances to all other profiles for every profile. As distance $\delta$ between two profiles Q and R we use the similarity measure $\Phi$ after converting it to a distance by subtracting it from 1 (cf., Sect. 2.3). We apply the similarity

measure to both profile pairs (Q, $R_w$) and ($Q_w$, R) to account for missing layers. Hence,

$$\delta(Q,R) = \max\{1 - \Phi(Q,R_w), \, 1 - \Phi(Q_w,R)\}. \tag{2}$$

The pairwise distances between the profiles of the group can be translated into a configuration plot, which gathers similar profiles close to each other, and dissimilar ones further away from each other[4] (Fig. 5). The medoid profile, being most similar to all other profiles, is the member of the group that represents the group the best. We use the same set of twelve profiles as in

Sect. 3.1 to demonstrate the profile aggregation. As the snowpack conditions within the set are too different to meaningfully represent them by one representative profile (Fig. 5a), we can combine clustering and aggregating to draw the representative profiles of the most distinct clusters within the set (Fig. 5b—the three clusters consist of profiles 1–4, 5–7, and 8–12, as depicted in Fig. 4).

While our proof of concept demonstrates a sound and reliable workflow, using the medoid profile may be computationally

too expensive to deal efficiently with data volumes beyond the order of tens to hundreds of profiles on an operational basis. However, Paparrizos and Gravano (2015) show that the medoid approach performs just better than any other sequence aggrega-

---

[4]We use an ordinal multi-dimensional scaling approach to create the configuration plot for the group of snow profiles (e.g., Mair, 2018).





tion method. *Dynamic Time Warping Barycenter Averaging* (Petitjean et al., 2011) might be an alternative aggregating method that could be evaluated in future studies.

## 4   Discussion and conclusions

The snow profile alignment algorithm and the similarity measure presented in this paper aim to address two of the main reasons

that have limited the adoption of snowpack models to support avalanche warning services and practitioners. First, the methods provide the foundation for numerical grouping and summarizing of snow profile data, and thus can help to make snowpack model output more accessible by addressing any avalanche operations' fundamental questions *Where in the terrain do we find which conditions?*. Second, our methods have the potential to make snowpack models more relevant to avalanche forecasters by providing a means for model evaluation against human observations.

Since we are the first to present an alignment algorithm and similarity measure for generic snow profiles, there is no objective approach to evaluate the methods against existing benchmark methods. Unlike in the data mining community, where distance measures are evaluated through classification applications (Wang et al., 2013), snow profile data sets that represent ground truth of alignments or groupings do not (yet) exist. Consequently, the evaluation of our methods needs to rely on expert judgement or application valuation. During the development of our algorithm, we therefore manually evaluated the alignment of many

profile pairs, a few of which are shown in the supplementary material (Sect. S3) to demonstrate its behavior. Furthermore, we evaluated the interplay of the alignment algorithm and similarity measure through clustering and aggregation applications. Those applications involve many individual profile assessments and therefore represent a meaningful valuation approach, which shows that the alignment algorithm and similarity measure are capable of distinguishing between subtle differences in the snow stratigraphy and yield a sound grouping that could have been carried out by a human avalanche professional.

Although our methods have been designed to accommodate a wide range of requirements and to cope with a variety of scenarios, the following limitations should be considered. First, while the layer matching algorithm can easily be applied to large data sets in an unsupervised manner, not all scenarios within such a data set might be best served with the same parameter settings. Getting meaningful results in a highly diverse data set requires the algorithm to be less constrained. However, this can also result in unrealistic alignments. Second, when alignments are based on grain type and hardness alone, the matching

of layers can sometimes be ambiguous—even for human experts. In those cases, labeling a few key layers with their burial date can greatly improve the alignment accuracy with little extra effort, especially as the labeling of weak layers is already established practise in North America (Canadian Avalanche Association, 2016). Note that our algorithm can easily be expanded to include other layer properties, especially if they are of ordinal or numeric data types such as e.g., grain size or specific surface area. And third, it is important to recognize that any numerical approach that condenses the complexity of these similarity

assessments to a one dimensional scale between $[0, 1]$ is unable to capture the full expertise and situational flexibility of human forecasters. However, it is a critical prerequisite for algorithmically grouping profiles and establishing ranks among them. As such our similarity measure offers a consistent evaluation of snow profiles that is based on the most commonly available layer characteristics.



**Figure 5.** Configuration plots of a group of twelve snow profiles, where similar profiles are gathered close to each other; (a) shows the geometric center of the whole group, which identifies the one, most representative profile; (b) shows the three most distinct subsets of the whole group and highlights their associated representative profiles, cf. Fig. 4.



Since our methods aim to offer direct benefit in helping to overcome the operational challenges of summarizing snowpack model data and evaluating that data, we imagine its integration into operational avalanche forecasting as follows. Traditionally, an avalanche risk management operation, such as a public warning service or a backcountry guiding operation, obtains vital information about the snowpack through manual snow pit observations at select point locations. Simulated snow profiles across

a mountain drainage can sample the snowpack conditions similarly to field observations, except with a higher spatiotemporal coverage and independent from external circumstances. Our methods could group the simulated profiles according to similar conditions, potentially uncovering different avalanche problems. Furthermore, that grouping could quantify the prevalence of those conditions, and the conditions could be linked to their specific location, elevation, and aspect. Then, the different conditions could be summarized by their representative profile to present the data in a familiar way. The human forecaster can use

that simulated data as additional data source complementing the field observations, or deploy targeted field observations to verify the model output. Through a continuous evaluation of the model output against human observations or human assessments (e.g., synthesized snow profiles), the momentary validity of the simulations could potentially be extrapolated into datasparse regions. More generally, a continuous evaluation of operational snowpack simulations provides an opportunity to better understand strengths and weaknesses of the involved model chain for its application in avalanche forecasting. Through this line of

research, we hope that snowpack models will be further incorporated into operational avalanche hazard assessment routines, so that avalanche forecasters can begin to build understanding in how to interpret and when to trust snowpack simulations.

*Code and data availability.* The snow profile alignment algorithm and similarity measure, as well as the data and the according analysis scripts to reproduce the results presented in this paper are available from an Open Science Framework registration. All code and data can be accessed through https://doi.org/10.17605/OSF.IO/9V8AD.

**Appendix A: An exemplary solution to the DTW optimization problem**

In this section we explain the concept of solving the DTW optimization problem and thereby finding the optimal warping path $P$ through the local cost matrix $\mathbf{D}$. We do that exemplarily by applying some of the same constraints that we also use in the snow profile alignment algorithm; most notably, those are the Sakoe-Chiba local slope constraint and (symmetric) open-end boundary conditions (cf., Sect. 2.2.4).

As mentioned in Sect. 2.2.1, the DTW optimization problem can be solved recursively with the aid of dynamic programming. Imagine a local cost matrix $\mathbf{D}$ ($i$-by-$j$) with individual elements $\mathrm{D}_{ij}$. Another, yet empty matrix $\mathbf{G}$—the *accumulated* cost matrix—has the same dimension as $\mathbf{D}$. From the boundary conditions, we know that the first items of the two sequences need to be matched. Thus the optimal warping path $P$ starts at $\mathrm{G}_{11}$, which holds the same value as $\mathrm{D}_{11}$. From the local slope constraint, we know that a horizontal or vertical step is only allowed if following a diagonal step. Therefore, as we are about

to do our first step, we have to do a diagonal one to $\mathrm{G}_{22}$. $\mathrm{G}_{22}$ is the second element of the optimal warping path $P$, and its value can be calculated by the accumulated cost one step before (i.e., $\mathrm{G}_{11}$) plus twice the *local* cost of the current step (i.e.,



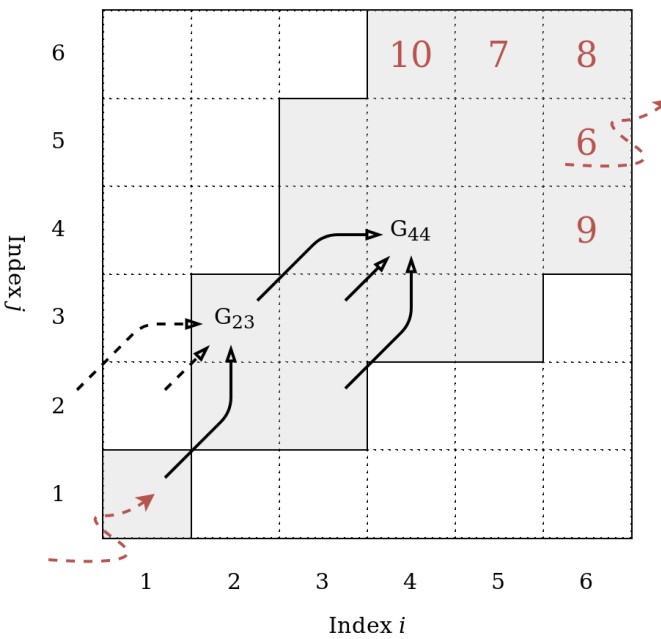

**Figure A1.** Sketch of a cost matrix $\mathbf{G}$ that stores the smallest accumulated cost of visiting a matrix element $G_{ij}$. In an open-end alignment, the optimal warping path $\boldsymbol{P}$ starts at $G_{11}$. The steps of the warping path are governed by the chosen local slope constraint (black arrows), such that only the grey matrix cells can be visited. The warping path ends in the last row or column of the matrix where the accumulated cost is smallest (i.e. at $G_{65} = 6$). See text for more details.

$D_{22}$); hence $G_{22} = G_{11} + 2D_{22}$ (where the weighting factor 2 indicates that *both* indices $i$ and $j$ have been incremented). As we just did a diagonal step, we are now allowed to step up vertically, diagonally, or horizontally to fields $G_{23}$, $G_{33}$, or $G_{32}$, respectively.

More generally, any element $G_{ij}$ can be calculated by the recursion

$$
5 \quad G_{ij} = \min \begin{bmatrix} G_{i-1\,j-2} + 2D_{i\,j-1} + D_{ij} \\ G_{i-1\,j-1} + 2D_{ij} \\ G_{i-2\,j-1} + 2D_{i-1\,j} + D_{ij} \end{bmatrix}. \tag{A1}
$$

Figure A1 sketches that concept. Each individual step of the optimal warping path $\boldsymbol{P}$ is governed by the local slope constraint, such that only a limited number of matrix elements can be visited by the warping path. Each of those elements of the cost matrix $\mathbf{G}$ in turn stores the *smallest* accumulated cost that is necessary to arrive at that cell. In a symmetric open-end alignment, the final element of the optimal warping path $\boldsymbol{P}$ is the one element in the last column or row of $\mathbf{G}$ that has accumulated the least cost.

If the optimal warping path $\boldsymbol{P}$ has $K$ elements from $k = 1, ..., K$, then each element $\boldsymbol{p}_k$ stores its location in the cost matrix by $\boldsymbol{p}_k = (p_k^i, p_k^j)$. Consequently, the accumulated cost of the optimal warping path $\boldsymbol{P}$ can be expressed as $G_{p_K^i\,p_K^j}$. For example, for the warping path implied by Fig. A1, $\boldsymbol{P} = \{(1,\,1), (2,\,2), ..., (6,\,5)\}$ and $G_{p_K^i\,p_K^j} = G_{65} = 6$. Finally, the DTW





distance between two sequences, $d_{\mathrm{DTW}}$, is expressed by the accumulated cost of the optimal warping path $\boldsymbol{P}$ normalized by the length of the path (expressed as Manhattan distance), i.e.

$$d_{\mathrm{DTW}} = \frac{1}{p_K^i + p_K^j}\, \mathrm{G}_{p_K^i\, p_K^j}. \tag{A2}$$

In Sect. 2.2.4 we introduced the *warped* snow profile $Q_{\mathrm{w}}$. With the insight gained from the current section, we can now
precisely define the warped profile $Q_{\mathrm{w}}$. Therefore we adopt the notation introduced in Sect. 2.2.3, and additionally denote the layer height of the snow profiles $Q_{\mathrm{w}}$ and R as $\boldsymbol{q}_{\mathrm{w}}{}^{\mathrm{Ht}}$ and $\boldsymbol{r}^{\mathrm{Ht}}$, respectively. Then the warped profile $Q_{\mathrm{w}}$ can be constructed with the indices $\boldsymbol{p^i}$ and $\boldsymbol{p^j}$, which are given by the warping path $\boldsymbol{P}$, by

$$\boldsymbol{q}_{\mathrm{w}}{}^{\mathrm{Ht}} = \boldsymbol{r}^{\mathrm{Ht}}{}_{\boldsymbol{p^j}}, \tag{A3}$$

$$\boldsymbol{q}_{\mathrm{w}}{}^{\mathrm{g}} = \boldsymbol{q}^{\mathrm{g}}{}_{\boldsymbol{p^i}}. \tag{A4}$$

The vectors $\boldsymbol{q}_{\mathrm{w}}{}^{h}$ and $\boldsymbol{q}_{\mathrm{w}}{}^{t}$ can be calculated analogously to Equ. (A4).

*Author contributions.* All authors conceptualized the research; SH provided the snowpack modeling infrastructure and the snowpack model output; FH derived the methods and implemented both code and simulations; all authors contributed to writing the manuscript; PH acquired the funding.

*Competing interests.* The authors declare that they have no conflict of interest.

*Acknowledgements.* FH thanks T. Giorgino for a valuable exchange on open-end DTW, as well as S. Mayer and B. Richter for exchanging research ideas and providing additional snow profile data for methods testing.





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
