# Peer review of "Snow profile alignment and similarity assessment for aggregating, clustering, and evaluating of snowpack model output for avalanche forecasting"

_Geoscientific Model Development, 2020_

## Referee Comment (RC1) · Anonymous Referee #1 · 14 Aug 2020

A motivated and well conducted application of dynamic time warping alignments

Note - this comment only covers the aspects concerning the alignment algorithm.

General comments —————-

In their paper, Herla et al. develop a model to compare snow packing profiles based on the alignment of their stratigraphy. The alignment procedure is based on the dynamic time warping (DTW) algorithm; in short, it enables the flexible comparison and clustering of snow layer profiles.

[Figure]

The paper is well written and it makes a convincing case for the use of DTW. Notably, it introduces non-trivial domain-specific adaptations: for one, a cost metric built from a weighted combination of heterogeneous discrete parameters (grain type, hardness, age). Each variable is preprocessed carefully in a knowledge-based way (e.g. grain type was addressed through a suitable similarity matrix). Another adaptation is the selection of open-ended alignments coupled with slope-limited patterns, with suitable weights and windows sizes serving as hyper-parameters. The description of the DTW algorithm and the construction of the local cost matrix D are appropriate and clear.

Specific and technical comments ——————

I only have a few minor comments, mostly for the sake of precision.

* Page 9. "In fig. 1..." . Figure 1 is illustrative, but it uses different warping parameters than your final set (e.g. constraints). Hence, consider adding "*As an illustration,* in Figure 1...".

* «no "stopping" or "going back" is allowed.» To be more precise: «no "going back in time" is allowed». Actually some patterns, such as your illustration of Figure 1, do allow local "stopping", as you note.

* In 2.2.4. (and possibly other places) "The symmetric Sakoe-Chiba local slope constraint" is mentioned. However Sakoe-Chiba's paper introduced *several* possible recursions (see their Table I). Specifically, the one you adopt appears to be "P=1, symmetric". This should be made explicit. You may also mention that it's "symmetricP1" in R.

* Please mention the dtw R package used and the corresponding version.

* "Mirroring D about its anti-diagonal". An alternative, possibly simpler way to express this is to say that one reverses both time series.

* Appendix A: "(the weighting factor 2 indicates that both indices i and j have been incremented)". Weighting factors, or slope weights, ensure that the normalisation prefactor in A2 is path-independent and has that specific form. [This is well explained in Rabiner-Juang's book on speech recognition.] Either omit the sentence or rephrase slightly, e.g., "the slope weights depend on the local indices' increments and ensure that alignments can be compared".

* "(expressed as Manhattan distance)" to clarify: "(expressed as Manhattan distance *from the matrix origin, for symmetric recursions*)"

* Possible material for SI: - A figure with step patterns and weights, e.g. subfigure A) plot(symmetric2) because it is used in Figure 1, and subfigure B) plot(symmetricP1), used by your alignment, for comparison.

TG

---

## Referee Comment (RC2) · Matthieu Lafaysse (Referee) · 15 Sep 2020

Summary and major comments

Herla et al. present a matching algorithm between snowpack profiles able to classify and evaluate snow cover simulations designed for avalanche hazard forecasting. I fully agree with the need of new tools to extend the evaluations of these models and facilitate their use by forecasters. The question is very well introduced in the paper. I am also glad to see that the proposal of Hagenmuller and Pilloix, 2016 to use such

maching algorithms for that purpose found an echo in other modelling teams and is applied in new contexts and with other similarity metrics. I also like the discussion about a possible concrete use of the method in an operational forecasting system. The results shown by the authors in a clustering application are interesting and a number of new advances in this paper are valuable for the community (new metrics, classification and synthesis methods).

However, I am a bit uncomfortable with the way the authors present their work in this context. Indeed their unique reference to Hagenmuller and Pilloix, 2016 is: "Hagenmuller and Pilloix, 2016 was the first to align, cluster, and aggregate one-dimensional snow hardness profiles from ram resistance field measurements using Dynamic Time Warping (DTW), a method from the fields of time series analysis and data mining. While their contribution demonstrates the usefullness of DTW for snow profile comparisons, their method is not general enough to allow for meaningful comparisons of snow profiles from different sources and varying levels of details." Then, they present their work as "a new approach for computationnally comparing, grouping and summarizing snow profiles" and they present the algorithm in details in Section 2.2.1 as it could be expected for the first publication of an algorithm. They also state in the discussion "we are the first to present an alignment algorithm and similarity measure for generic snow profiles". However, the method proposed in this paper can not be told as new. Indeed, the algorithm is extremely similar to the one developed by our colleague P. Hagenmuller and which is already used in several publications (with or without a detailed description of DTW depending on the references): Hagenmuller and Pilloix, 2016; Teich et al., 2019; Hagenmuller et al. 2018a; Hagenmuller et al. 2018b; Viallon et al. (accepted). Note also that DTW was also already applied on snow by Schaller et al., 2016. The innovation of Herla et al. is mainly the distance the authors introduce in the application of this algorithm, in order to use a criteria more focused on mechanical stability, but the general philosophy of the method is the same. Furthermore, the application of such algorithm for clustering also already appears in Hagenmuller et al., 2018b and also include a selection method of a representative profile. Although this reference is not

a publication in a peer-review journal but a conference proceedings, the authors are aware of this contribution.

I think there is room for everyone to work on similar topics if the context is fairly presented. Therefore I think that the authors should consider the following modifications and questions:

- The introduction should better emphasize the own innovations of this paper and avoid general statements presenting the whole method as a new algorithm.

- Section 2.2.1 should better emphasize what is common and what differs between this algorithm and the algorithm of Hagenmuller and Pilloix, 2016 and following papers. The details of DTW already descibed in previous literature can be moved to an Appendix in order to be more focused on the novelties in the main text.

- Then, it is unclear what is unsatisfying in the algorithm of P. Hagenmuller. What do the author mean by "not general enough to allow for meaningful comparisons of snow profiles"? Hagenmuller and Pilloix, 2016, discuss the possibility to modify the distance criteria depending on applications: "The metrics D and V between profiles whose definition involve the mean square difference of logarithmic hardness can be adapted to incorporate other snow properties. [...]" with examples. In Viallon et al., accepted, we applied DTW with a more general distance combining density, liquid water content, grain shape and depth, better suited for an overall model evaluation and the algorithm behaves well in this context. The distance presented here by the authors is probably better suited to their application but can not pretend to be more "general". Do the authors refer only to the management of missing values in snow profiles? Or is there something else? I think it is important to be more specific on the issues in previous references to better justify and emphasize the innovations they want to publish here.

- A number of choices in the distance definition are model-dependent relatively to the SNOWPACK model. For instance, neither layer hardness or layer date are diagnostics

of the Crocus model (similar to SNOWPACK in terms of complexity). Furthermore, a number of considerations in the distance definition for grain type are based on considerations about the current typical behaviour of SNOWPACK (lines 5-6 and 23-31 page 5). These considerations might not apply to another model or even to a future version of SNOWPACK with for instance new parameterizations of snow metamorphism. This option can be arguable if there is added value to make this choice but the limitation should be clearly discussed. Will it be necessary to modify the distance definition if significant changes are implemented in the model?

- Finally, it is questionable whereas there is really added value for the community in the future to provide two separate codes from two research teams based on a similar algorithm. The authors requested the code of P. Hagenmuller last winter and they received it with a number of explanations from P. Hagenmuller. This is surprisingly not mentioned in the Acknowledgements section. I think the authors should justify in the paper the need for another code by describing the reasons which have probably prevent them using directly the code from P. Hagenmuller. A better understanding of these limitations might help to avoid further work duplication and hopefully allow more shared developments in the post-processing of snow models in the future, although I am fully aware of the difficulties of such collaborative developments.

Other comments

Page 10 Line 15: Is there a limitation of snow heights differences to apply the rescaling? Is it meaningful to rescale profiles even when the relative difference of snow height reach for instance 500% or more?

Page 10 Line 21: The formulation is a bit ambiguous. Is the value of 0.5 cm was indeed chosen for grid resampling in this work?

Page 15 Lines 27-28 The variables chosen to compute the similarity measure are not prognostic variables of the model but rather indirect diagnostics (grain type, hardness) involving very uncertain parameterizations from prognostic variables. Beyond the clustering application presented in this paper, the possible use of this criteria for model evaluation raises some questions. Indeed, it can not be established if this metric would be really representative of the real skill of the physical model or if these transformation functions would not prevail in the obtained metric. I fully understand the motivation relative to the avalanche hazard application but this topic should be discussed because (1) a perfect model in terms of physical evolution laws might not provide a perfect similarity measure due to errors in the variables transformation and (2) this criteria might not be recommanded for a process-related model evaluation where I think a metric based on density and temperature profiles for instance should be preferred. In spite of this comment, I acknowledge that the detailed thinking about an appropriate weighting of layers in the final similarity metric is really interesting and useful (although a bit complex) from the mechanical point of view.

Competing interest

I mention as competing interest that I work in the same research group as Pascal Hagenmuller.

References

Hagenmuller, P. and T. Pilloix (2016): A new method for comparing and matching snow profiles, application for profiles measured by penetrometers. Frontiers in Earth Science, 4 , 52, https://doi.org/10/3389/feart.2016.00052.

Teich, M., A. D. Giunta, P. Hagenmuller, P. Bebi, M. Schneebeli and M. J. Jenkins (2019). Effects of bark beetle attacks on forest snowpack and avalanche formation – implications for protection forest management. Forest Ecology and Management, 438, 186-203, https://doi.org/10.1016/j.foreco.2019.01.052

Hagenmuller P., A. van Herwijnen, C. Pielmeier, H.-P. Marshall (2018a). Evaluation of the Snow Penetrometer Avatech SP2. Cold Regions Science and Technology, 149, 83-94, https://doi.org/10.1016/j.coldregions.2018.02.006

Hagenmuller, P., L. Viallon, C. Bouchayer, M. Teich, M. Lafaysse and V. Vionnet, (2018b) Quantitative comparison of snow profiles. International Snow Science Workshop Proceedings 2018, Innsbruck, Austria, 876-879,https://arc.lib.montana.edu/snowscience/objects/ISSW2018_O10.5.pdf

Schaller, C. F., Freitag, J., Kipfstuhl, S., Laepple, T., Steen-Larsen, H. C., and Eisen, O. (2016): A representative density profile of the North Greenland snowpack, The Cryosphere, 10, 1991–2002, https://doi.org/10.5194/tc-10-1991-2016, 2016.

Viallon, L., Hagenmuller, P., Lafaysse, M.: Forcing and evaluating detailed snow cover models with stratigraphy observations, accepted in Cold Reg. Sci. Technol.

---

## Author Comment (AC1) · 28 Oct 2020

October 28, 2020

**Responses to Referee #1 (TG)**

General Comment

Referee General Comment: *In their paper, Herla et al. develop a model to compare snow packing profiles based on the alignment of their stratigraphy. The alignment procedure is based on the dynamic time warping (DTW) algorithm; in short, it enables the flexible comparison and clustering of snow layer profiles. The paper is well written and it makes a convincing case for the use of DTW. Notably, it introduces non-trivial domain-specific adaptations: for one, a cost metric built from a weighted combination of heterogeneous discrete parameters (grain type, hardness, age). Each variable is preprocessed carefully in a knowledge-based way (e.g. grain type was addressed through a suitable similarity matrix). Another adaptation is the selection of open-ended alignments coupled with slope-limited patterns, with suitable weights and windows sizes serving as hyper-parameters. The description of the DTW algorithm and the construction of the local cost matrix D are appropriate and clear.*

**Author General Comment**: We thank TG for his constructive review and helpful comments. We appreciate the encouraging comments about our domain-specific adaptations of DTW. Please see below for our responses to specific comments and suggestions from Referee #1. Additions to the manuscript are included in our responses in quotes, where page- and line-numbers refer to the revised manuscript that includes the highlighted and marked-up changes.

Specific Comments

Referee Comment (RC) 1.1: *Page 9. "In fig. 1..." . Figure 1 is illustrative, but it uses different warping parameters than your final set (e.g. constraints). Hence, consider adding "\*As an illustration,\* in Figure 1...".*

**Author Response (AR) 1.1:** Thank you! Done. (P10 L16)

RC 1.2: *'no "stopping" or "going back" is allowed.' To be more precise: 'no "going back in time" is allowed'. Actually some patterns, such as your illustration of Figure 1, do allow local "stopping", as you note.*

**AR 1.2:** Thank you! Changed it. (P9 L16)

RC 1.3: *In 2.2.4. (and possibly other places) "The symmetric Sakoe-Chiba local slope constraint" is mentioned. However Sakoe-Chiba's paper introduced \*several\* possible recursions (see their Table I). Specifically, the one you adopt appears to be "P=1, symmetric". This should be made explicit. You may also mention that it's "symmetricP1" in R.*

**AR 1.3:** We agree and made this more explicit. (P14 L9ff, P24 L4)

RC 1.4: *Please mention the dtw R package used and the corresponding version.*

**AR 1.4:** Done. We added a brief note in the Discussion (P22 L5) and a more detailed explanation to the Code Availability section (P23 L28).

RC 1.5: *"Mirroring D about its anti-diagonal". An alternative, possibly simpler way to express this is to say that one reverses both time series.*

**AR 1.5:** Good suggestion! Changed. (P14 L26)

RC 1.6: *Appendix A: "(the weighting factor 2 indicates that both indices i and j have been incremented)". Weighting factors, or slope weights, ensure that the normalisation prefactor in A2 is path-independent and has that specific form. [This is well explained in Rabiner-Juang's book on speech recognition.] Either omit the sentence or rephrase slightly, e.g., "the slope weights depend on the local indices' increments and ensure that alignments can be compared".*

**AR 1.6:** Thanks, we rephrased the explanation. (P24 L13f)

RC 1.7: *"(expressed as Manhattan distance)" to clarify: "(expressed as Manhattan distance \*from the matrix origin, for symmetric recursions\*)"*

**AR 1.7:** Done. (P25 L14)

RC 1.8: *Possible material for SI: - A figure with step patterns and weights, e.g. subfigure A) plot(symmetric2) because it is used in Figure 1, and subfigure B) plot(symmetricP1), used by your alignment, for comparison.*

**AR 1.8:** Thanks for the suggestion. We included the figure and its explanation in an Appendix B. (P25 L23ff)

**Responses to Referee #2 (Matthieu Lafaysse)**

General Comment

Referee General Comment RC 2.1: *Herla et al. present a matching algorithm between snowpack profiles able to classify and evaluate snow cover simulations designed for avalanche hazard forecasting. I fully agree with the need of new tools to extend the evaluations of these models and facilitate their use by forecasters. The question is very well introduced in the paper. I am also glad to see that the proposal of Hagenmuller and Pilloix, 2016 to use such matching algorithms for that purpose found an echo in other modelling teams and is applied in new contexts and with other similarity metrics. I also like the discussion about a possible concrete use of the method in an operational forecasting system. The results shown by the authors in a clustering application are interesting and a number of new advances in this paper are valuable for the community (new metrics, classification and synthesis methods).*

*However, I am a bit uncomfortable with the way the authors present their work in this context. Indeed their unique reference to Hagenmuller and Pilloix, 2016 is: "Hagenmuller and Pilloix (2016) and Hagenmuller (2018) were the first to align, cluster and aggregate one-dimensional snow hardness profiles from ram resistance field measurements using Dynamic Time Warping (DTW), a method from the fields of time series analysis and data mining. While their contribution demonstrates the usefulness of DTW for snow profile comparisons, their method is not general enough to allow for meaningful comparisons of snow profiles from different sources and varying levels of details." Then, they present their work as "a new approach for computationally comparing, grouping and summarizing snow profiles" and they present the algorithm in details in Section 2.2.1 as it could be expected for the first publication of an algorithm. They also state in the discussion "we are the first to present an alignment algorithm and similarity measure for generic snow profiles". However, the method proposed in this*
[Figure]

*paper can not be told as new. Indeed, the algorithm is extremely similar to the one developed by our colleague P. Hagenmuller and which is already used in several publications (with or without a detailed description of DTW depending on the references): Hagenmuller and Pilloix, 2016; Teich et al.,2019; Hagenmuller et al. 2018a; Hagenmuller et al. 2018b; Viallon et al. (accepted). Note also that DTW was also already applied on snow by Schaller et al., 2016. The innovation of Herla et al. is mainly the distance the authors introduce in the application of this algorithm, in order to use a criteria more focused on mechanical stability, but the general philosophy of the method is the same. Furthermore, the application of such algorithm for clustering also already appears in Hagenmuller et al., 2018b and also include a selection method of a representative profile. Although this reference is not a publication in a peer-review journal but a conference proceedings, the authors are aware of this contribution.*

**Author General Comment AC 2.1:** We thank Matthieu Lafaysse for his review, in particular for the encouraging comment related to our attempt of creating tools of operational value for avalanche forecasting, as well as for the honest presentation of his concerns. In his review we identified three main overarching themes that we will address first before responding to the more technical comments in a point-by-point manner further below.

The main concerns of the referee are centered around proper accreditation, clear communication of our innovations as well as clear separation towards other contributions. We addressed these issues altering the Introduction, the Discussion, and the Abstract, including the additional references suggested by the referee, highlighting how we identified research needs, and more explicitly communicating the innovations of our contribution. Our main modifications include the following (new additions and edits are printed in black fonts, while text from our first submission is printed in gray; page- and line-numbers refer to the revised manuscript with the marked-up changes):

- We added an explanation of the data type we are working with:

  *"When observing a snow profile, traditionally the most commonly recorded layer characteristics are snow grain type, grain size and layer hardness. [...] **Snow profiles that contain information about these layer characteristics are referred to as** generic *snow profiles hereafter and represent the main source of snow stratigraphy information in operational contexts."* (P2 L15ff)

- We provide a more detailed summary of how DTW has been introduced to the snow community, and we illustrate in more detail how the layer matching algorithm by Hagenmuller and Pilloix (2016) has been applied in the snow community. Furthermore, we made the research gap more specific:

  *"Hagenmuller and Pilloix (2016) and **Hagenmuller et al. (2018b), as well as Schaller et al. (2016)** introduced Dynamic Time Warping (DTW), a longstanding method from the fields of time series analysis and data mining, to the snow community. **Both implemented a layer matching algorithm to align, cluster, and aggregate one-dimensional snow hardness (or density) profiles from field measurements and thereby demonstrated the usefulness of DTW for snow profile comparisons. In the subsequent years, the layer matching algorithm by Hagenmuller and Pilloix (2016) has been applied to evaluate snow penetrometers (Hagenmuller et al., 2018a) or to characterize spatial variability of the snow cover from ram resistance field measurements (Teich et al., 2019). Consequently, their approach has focused on one-dimensional, continuous, numerical sequences and is not readily applicable to operational snowpack observations by avalanche forecasters."* (P4 L4ff)

- We then explicitly formulate our research goals:

  *"The objective of this study is to introduce an approach for computationally comparing, grouping and summarizing generic snow profiles **that consist of multidimensional, discrete sequences of categorical, numerical, and ordinal data types.** To maximize the value for avalanche forecasting, our methods focus on

structural elements in the profiles that are particularly important for avalanche
hazard assessments and can handle both simulated profiles and manual obser-
vations with different levels of detail." (P4 L14ff)

- We re-iterate the innovations of our contribution in the Discussion, where we
  also include the recently published paper by the referee's group (Viallon-Galinier
  et al., 2020). Since the efforts behind that paper were not published or otherwise
  publicly accessible until mid to end September 2020, we could not have included
  them in our original submission.

*"Building on the well-established and longstanding concept of Dynamic Time
Warping (DTW), we developed a snow profile alignment algorithm that combines
multiple layer characteristics of categorical, numerical, and ordinal format into a
weighted metric and feeds into existing DTW algorithms such as the open-source
R package* dtw. *Moreover, we reviewed and derived useful DTW configurations
and hyper-parameter settings for snow profile applications. Since these appli-
cations rely on operationally available profile observations that typically focus on
information relevant for current avalanche conditions only, our approach is able to
handle missing data and take advantage of select layer date tags. To maximize
the layer matching performance for profiles with limited details, we implemented
a scheme for preferential layer matching based on domain knowledge. In par-
allel, Viallon-Galinier et al. (2020) extended the layer matching algorithm from
Hagenmuller and Pilloix (2016) to conduct a detailed, process-related evaluation
of the snowpack model Crocus based on high-quality snow profile observations
with a large variety of observed variables that are sampled at specific study sites
at regular intervals. Since their goal is the correction of deviating model states
with a direct insertion assimilation scheme based on point scale simulations and
observations, their evaluation targets not only each individual layer, but also each
individual layer characteristic separately. To address operational avalanche fore-*

*casting needs, we additionally developed a similarity measure that focuses on
avalanche forecasting specific considerations where certain layers are consid-
ered more important. Moreover, combining information from individual layers and
their characteristics into a scalar measure allows for clustering and aggregating of
sets of profiles to characterize and evaluate the regional scale avalanche hazard
conditions."* (P22 L1ff)

Specific Comments

RC 2.2: *The introduction should better emphasize the own innovations of this paper
and avoid general statements presenting the whole method as a new algorithm.*

**AR 2.2:** Done, see general comment above. Additionally, we removed all ambiguous
statements.

RC 2.3: *Section 2.2.1 should better emphasize what is common and what differs be-
tween this algorithm and the algorithm of Hagenmuller and Pilloix, 2016 and following
papers. The details of DTW already described in previous literature can be moved to
an Appendix in order to be more focused on the novelties in the main text.*

**AR 2.3:** We improved upon the distinction between the different contributions, see
general comment above. However, we prefer not to move the background section
on DTW (i.e., Section 2.2.1) to an appendix. First, we believe it is appropriate and
necessary for a methods paper to transparently communicate the presented method.
Second, in our opinion, other publications in the snow community that present or use
a matching algorithm based on DTW have so far not presented the method in an
accessible and reproducible manner. And third, the journal addresses a wider audi-
ence of geophysicists to whom a matching algorithm for multi-dimensional, mixed-type
sequences or time series might be of relevance. Demonstrating the modification and
application of DTW based on an illustrative example such as snow profiles, while

relating to the rich body of literature in the DTW community, will allow readers of gmd to develop similar methods for other geophysical disciplines.

RC 2.4: *Then, it is unclear what is unsatisfying in the algorithm of P. Hagenmuller. What do the author mean by "not general enough to allow for meaningful comparisons of snow profiles"? Hagenmuller and Pilloix, 2016, discuss the possibility to modify the distance criteria depending on applications: "The metrics D and V between profiles whose definition involve the mean square difference of logarithmic hardness can be adapted to incorporate other snow properties. [...]" with examples. In Viallon et al., accepted, we applied DTW with a more general distance combining density, liquid water content, grain shape and depth, better suited for an overall model evaluation and the algorithm behaves well in this context. The distance presented here by the authors is probably better suited to their application but can not pretend to be more "general". Do the authors refer only to the management of missing values in snow profiles? Or is there something else? I think it is important to be more specific on the issues in previous references to better justify and emphasize the innovations they want to publish here.*

**AR 2.4:** We addressed this concern in the modifications described in the general comment above.

RC 2.5: *A number of choices in the distance definition are model-dependent relatively to the SNOWPACK model. For instance, neither layer hardness or layer date are diagnostics of the Crocus model (similar to SNOWPACK in terms of complexity). Furthermore, a number of considerations in the distance definition for grain type are based on considerations about the current typical behaviour of SNOWPACK (lines 5-6 and 23-31 page 5). These considerations might not apply to another model or even to a future version of SNOWPACK with for instance new parameterizations of snow metamorphism. This option can be arguable if there is added value to make this choice but*

[Figure]

*the limitation should be clearly discussed. Will it be necessary to modify the distance definition if significant changes are implemented in the model?*

**AR 2.5:** We appreciate that the referee highlighted the close tie of our algorithm to the SNOWPACK model and is interested in limitations with respect to future version of SNOWPACK and their parametrizations of snow metamorphism.

To explain how the simulated variables are calculated and to explain the relation of our algorithm to SNOWPACK, we added the following paragraph at the beginning of Section 2.1: *"While grain types are computed by snow models based on parametrizations of snow metamorphism, and simulated burial date information can easily be derived from simulated deposition date or age of the layer, layer hardness is only a diagnostic variable provided by the model* SNOWPACK*, but not by Crocus. Therefore the following distance functions are presented in the light of* SNOWPACK*, and the application to other snow model output may require some modifications."* (P5 L8ff) More details about the conversion of layer age or date information is then provided in Sect. 2.1.3.

To communicate our reasoning behind the modifications of grain type similarities more clearly, we rewrote the following paragraph: *"SH and DH layers are formed by very different processes—by the deposition of hoar onto the snow surface versus by kinetic growth of crystals within the snowpack. Consequently, Lehning et al. (2001) evaluated the similarity of the two grain types as completely dissimilar. However, both SH and DH represent hazardous weak layers and are of comparable importance in avalanche hazard assessments (Schweizer and Jamieson, 2001). Furthermore, practical experience with the current version of* SNOWPACK *shows that SH layers are often converted to DH layers once buried. To account for both of these aspects, we raised their similarity from* 0 *to* 0.9 *for both tasks (Table 1A, B). "* (P7 L1ff)

To summarize: While our modifications to the grain type similarities accommodate peculiarities of SNOWPACK, they are all rooted in considerations of avalanche hazard assessment. We do therefore not expect any necessary modifications for

future versions of snowpack that include new parametrizations of snow metamorphism.

RC 2.6: *Finally, it is questionable whereas there is really added value for the community in the future to provide two separate codes from two research teams based on a similar algorithm. The authors requested the code of P. Hagenmuller last winter and they received it with a number of explanations from P. Hagenmuller. This is surprisingly not mentioned in the Acknowledgements section. I think the authors should justify in the paper the need for another code by describing the reasons which have probably prevent them using directly the code from P. Hagenmuller. A better understanding of these limitations might help to avoid further work duplication and hopefully allow more shared developments in the post-processing of snow models in the future, although I am fully aware of the difficulties of such collaborative developments.*

**AR 2.6:** We apologize for this oversight and we now thank P. Hagenmuller in the Acknowledgements for an email exchange at the outset of this research project. Furthermore, we agree that it is desirable to reduce redundant work and combine efforts, especially in a niche research field like snowpack modeling and avalanche forecasting. However, we also believe that there is value in developing multiple approaches and letting the research community decide what works best for their specific objectives. Furthermore, we strongly believe that the open access approach is most valuable for the research community and therefore built our algorithm on the open source R package dtw, which provides excellent documentation and is also implemented in other programming languages, instead of Hagenmuller's code.

RC 2.7: *Is there a limitation of snow height differences to apply the rescaling? Is it meaningful to rescale profiles even when the relative difference of snow height reach for instance 500% or more?*

**AR 2.7:** If the two profiles stem from the same forecast area, or in other words from

an area with homogeneous (albeit elevation and aspect dependent) meteorological conditions, rescaling of the profiles is reasonable independently from the relative difference of the snow heights. Especially when the relative difference of snow heights is extremely large, the two profiles have been subject to systematic variations in their meteorological forcing (e.g., increase of snowfall amounts with elevation). It is exactly these situations, where the rescaling is of particular importance.

Since that question is also related to clustering and aggregating applications, we added the following clarification to the Discussion. *"The similarity measure, and consequently the clustering and aggregating applications are purely based on the agreement of the snowpack structure. Hence, snow depth is not a driver of our similarity assessment unless it leads to deviations in the snow stratigraphy. If combined with monitoring of the snow depths distribution, a clustering or aggregation application can provide a comprehensive picture of the conditions within a specific forecast area."* (P23 L6ff)

RC 2.8: *Page 10 Line 21: The formulation is a bit ambiguous. Was the value of 0.5 cm indeed chosen for grid resampling in this work?*

**AR 2.8:** We added an explanation, so that the relevant paragraph reads: *"Once the profiles have been rescaled, each of the two profiles consists of a series of discrete layers along an irregular height grid. To equalize the two different height grids, we resample the profiles onto a regular grid with a constant sampling rate, which represents the final resolution for the alignment procedure. While our algorithm allows users to flexibly set the sampling rate, a resolution of about half a centimeter ensures that typically thin, hazardous weak layers are being captured. Hence, snow profiles in the presented examples have been resampled to $0.5$ cm."* (P11 L15ff)

RC 2.9: *The variables chosen to compute the similarity measure are not prognostic variables of the model but rather indirect diagnostics (grain type, hardness) involving*

*very uncertain parameterizations from prognostic variables. Beyond the clustering application presented in this paper, the possible use of this criteria for model evaluation raises some questions. Indeed, it can not be established if this metric would be really representative of the real skill of the physical model or if these transformation functions would not prevail in the obtained metric. I fully understand the motivation relative to the avalanche hazard application but this topic should be discussed because (1) a perfect model in terms of physical evolution laws might not provide a perfect similarity measure due to errors in the variables transformation and (2) this criteria might not be recommended for a process-related model evaluation where I think a metric based on density and temperature profiles for instance should be preferred. In spite of this comment, I acknowledge that the detailed thinking about an appropriate weighting of layers in the final similarity metric is really interesting and useful (although a bit complex) from the mechanical point of view.*

**AR 2.9:** We agree with the referee in that a model evaluation based on indirect diagnostics skews the obtained results and does not provide a direct evaluation of the physical model skill. Since our research focuses on making snowpack simulations more accessible for avalanche forecasters, and since those indirect diagnostics are the relevant variables for avalanche hazard applications, an evaluation limited to the prognostic variables is not informative and relevant in our context. To convey this idea more strongly, we extended the relevant paragraph in the Introduction:

*"Furthermore, the ability to operationally compare simulated snow profiles against observed ones provides an avenue for continuously monitoring the quality of the simulations and correct them if necessary. To judge the operational value of snowpack models for avalanche forecasting, it is particularly important to focus on snowpack features and layer characteristics that are of direct relevance for avalanche hazard assessments. Since operational snowpack observations and relevant layer characteristics are expressed by variables (such as grain type and layer hardness) that are only indirectly diagnosed by models, the parametrization from prognostic variables introduces another*

*layer of uncertainty. The evaluation of these models for practical purposes therefore needs to take all of these uncertainties into account.*" (P3 L26ff)

**References**

Hagenmuller, P. and T. Pilloix, 2016: A New Method for Comparing and Matching Snow Profiles, Application for Profiles Measured by Penetrometers. *Frontiers in Earth Science*, **4**, doi:10. 3389/feart.2016.00052.

Hagenmuller, P., A. van Herwijnen, C. Pielmeier, and H.-P. Marshall, 2018a: Evaluation of the snow penetrometer Avatech SP2. *Cold Regions Science and Technology*, **149**, 83–94, doi: 10.1016/j.coldregions.2018.02.006.

Hagenmuller, P., L. Viallon, C. Bouchayer, M. Teich, M. Lafaysse, and V. Vionnet, 2018b: Quantitative Comparison of Snow Profiles. *Proceedings of the 2018 international snow science workshop, Innsbruck, AUT*, 876—-879, https://arc.lib.montana.edu/snow-science/item/2668.

Lehning, M., C. Fierz, and C. Lundy, 2001: An objective snow profile comparison method and its application to SNOWPACK. *Cold Regions Science and Technology*, **33 (2-3)**, 253–261, doi:10.1016/s0165-232x(01)00044-1.

Schaller, C. F., J. Freitag, S. Kipfstuhl, T. Laepple, H. Christian Steen-Larsen, and O. Eisen, 2016: A representative density profile of the North Greenland snowpack. *Cryosphere*, **10 (5)**, 1991–2002, doi:10.5194/tc-10-1991-2016, https://tc.copernicus.org/articles/10/1991/2016/.

Schweizer, J. and J. B. Jamieson, 2001: Snow cover properties for skier triggering of avalanches. *Cold Regions Science and Technology*, **33 (2-3)**, 207–221, doi:10.1016/ S0165-232X(01)00039-8.

Teich, M., A. D. Giunta, P. Hagenmuller, P. Bebi, M. Schneebeli, and M. J. Jenkins, 2019: Effects of bark beetle attacks on forest snowpack and avalanche formation — Implications for protection forest management. *Forest Ecology and Management*, **438**, 186–203, doi: 10.1016/j.foreco.2019.01.052.

Viallon-Galinier, L., P. Hagenmuller, and M. Lafaysse, 2020: Forcing and evaluating detailed snow cover models with stratigraphy observations. *Cold Regions Science and Technology*, **180**, 103 163, doi:10.1016/j.coldregions.2020.103163.